# Stalled response near thermal equilibrium in periodically driven systems

Lennart Dabelow [1,3] & Peter Reimann [2] ✉

The question of how systems respond to perturbations is ubiquitous in physics. Predicting this response for large classes of systems becomes particularly challenging if many degrees of freedom are involved and linear response theory cannot be applied. Here, we consider isolated many-body quantum systems which either start out far from equilibrium and then thermalize, or find themselves near thermal equilibrium from the outset. We show that time-periodic perturbations of moderate strength, in the sense that they do not heat up the system too quickly, give rise to the following phenomenon of stalled response: While the driving usually causes quite considerable reactions as long as the unperturbed system is far from equilibrium, the driving effects are strongly suppressed when the unperturbed system approaches thermal equilibrium. Likewise, for systems prepared near thermal equilibrium, the response to the driving is barely noticeable right from the beginning. Numerical results are complemented by a quantitatively accurate analytical description and by simple qualitative arguments.

Understanding the effect of time-dependent perturbations on many-body quantum systems is a fundamental problem of immediate practical relevance. Examples include the implementation of cold-atom[1–6] and polarization-echo[6–8] experiments, or the control of general-purpose quantum computers and simulators[2,3,6,9]. Periodic driving, in particular, has been exploited to design so-called time crystals[10] and various metamaterials with unforeseen topological and dynamical properties, whose exploration has only just begun[11–14].

In this context, the majority of previous studies focused on the long-time behavior and, in particular, on the properties of the so-called Floquet Hamiltonian. A key aspect of such an approach is that it can only capture the actual behavior of the periodically driven system stroboscopically in time, i.e., at integer multiples of the driving period, whereas the possibly still very rich behavior in between those discrete time points remains inaccessible. For instance, the stroboscopic dynamics may appear nearly stationary even though the full, continuous dynamics still exhibits oscillations with large amplitudes.

We adopt a complementary perspective and explore the continuously time-resolved response on short-to-intermediate time scales.

Intuitively, one might naturally expect that periodic forcing leads to a clearly noticeable change of the observable properties if its strength and period are of the same order as the main intrinsic energy and time scales of the undriven system.

In this work, we show that such a fairly pronounced response is indeed observed for isolated many-body systems that are far away from thermal equilibrium. Our main discovery, however, is that this intuitively expected response is strongly suppressed near thermal equilibrium, at least as long as heating effects of the driving remain negligible. We dub this phenomenon "stalled response" in view of its two principal manifestations: For a system that is prepared far away from equilibrium, the observable response dies out as soon as the corresponding undriven reference system approaches thermal equilibrium. Similarly, when the system already starts out in thermal equilibrium, the driving is barely noticeable right from the beginning. In both cases, it is only at much later times that the driving effects may reappear in the form of very slow heating. Besides numerical evidence from several examples, we support our general prediction of stalled response near thermal equilibrium with simple heuristic arguments and with an analytical theory for large classes of many-body systems. Remarkably, we can also identify the main qualitative signatures of

[1]RIKEN Center for Emergent Matter Science (CEMS), Wako, Saitama 351-0198, Japan. [2]Faculty of Physics, Bielefeld University, 33615 Bielefeld, Germany. [3]Present address: School of Mathematical Sciences, Queen Mary University of London, London E1 4NS, UK. ✉e-mail: reimann@physik.uni-bielefeld.de

such a stalled response behavior in data from a very recent NMR experiment[8].

## Results

We consider periodically driven many-body systems with Hamiltonians

$$H(t) = H_0 + f(t)\,V\,,\tag{1}$$

where $H_0$ models some unperturbed reference system, $V$ is a perturbation operator, and $f(t) = f(t + T)$ is a (scalar) function with period $T$.

As usual, the expectation value of an observable (Hermitian operator) $A$ then follows as

$$\langle A \rangle_{\rho(t)} := \mathrm{tr}\,\{\rho(t)A\}\,,\tag{2}$$

where $\rho(t) := \mathcal{U}(t)\rho(0)\mathcal{U}^\dagger(t)$ is the (pure or mixed) system state at time $t$ if the initial condition was $\rho(0)$, and the propagator $\mathcal{U}(t)$ satisfies $\frac{d}{dt}\mathcal{U}(t) = -iH(t)\mathcal{U}(t)$ and $\mathcal{U}(0) = \mathbb{1}$ (identity operator). Likewise, the unperturbed system starts out from the same initial state $\rho(0)$, and then evolves into $\rho_0(t)$ under the time-independent Hamiltonian $H_0$, yielding expectation values $\langle A \rangle_{\rho_0(t)} := \mathrm{tr}\,\{\rho_0(t)A\}$. Accordingly, the system's response to the driving is monitored by the deviations of $\langle A \rangle_{\rho(t)}$ from $\langle A \rangle_{\rho_0(t)}$.

### Phenomenology

To illustrate the announced phenomenon of stalled response, we first present a numerical example in Fig. 1. Its specific choice is mainly motivated by the fact that it will admit a direct comparison with our analytical theory (presented below) without any free fit parameter. Further examples will be provided later.

As sketched in Fig. 1, we consider an $L \times L$ spin-$\frac{1}{2}$ lattice with $L = 5$ and open boundary conditions, where nearest neighbors are coupled by Heisenberg terms in the unperturbed system (solid links in the sketch),

$$H_0 := \sum_{i,j=1}^{L-1} \boldsymbol{\sigma}_{i,j} \cdot (\boldsymbol{\sigma}_{i+1,j} + \boldsymbol{\sigma}_{i,j+1})\,.\tag{3}$$

The vector $\boldsymbol{\sigma}_{i,j} = (\sigma_{i,j}^x, \sigma_{i,j}^y, \sigma_{i,j}^z)$ collects the Pauli matrices acting on site $(i,j)$. The perturbation additionally introduces spin-flip terms in the $z$

direction between next-nearest neighbors (dashed links in the sketch),

$$V := \sum_{i,j=1}^{L-1} \sum_{\alpha=x,y} (\sigma_{i,j}^\alpha \sigma_{i+1,j+1}^\alpha + \sigma_{i+1,j}^\alpha \sigma_{i,j+1}^\alpha)\,.\tag{4}$$

Since the magnetization $S^z := \sum_{i,j} \sigma_{i,j}^z$ commutes with both $H_0$ and $V$, we focus on one of the two largest subsectors, namely the one with eigenvalue $-1$ for $S^z$.

To prepare the system out of equilibrium, we fix the spins at sites $(2, 2)$ and $(3, 3)$ in the "up" state (red in the sketch in Fig. 1) and orient all other spins randomly. To obtain a well-defined energy, we additionally emulate a macroscopic energy measurement by acting with a Gaussian filter[15–17] of a target mean energy $E = -12$ and standard deviation $\Delta E = 4$ on the so-defined state. Formally, the initial condition can thus be expressed as $\rho(0) = |\psi\rangle\langle\psi|$ with

$$|\psi\rangle \propto e^{-(H_0-E)^2/4\Delta E^2}\, Q\,|\phi\rangle\,,\tag{5}$$

where $|\phi\rangle$ is a Haar-random state in the $S^z = -1$ sector. The projector $Q := \pi_{2,2}^+ \pi_{3,3}^+$ with $\pi_{i,j}^\pm := (1 \pm \sigma_{i,j}^z)/2$ enforces $\sigma_{2,2}^z = \sigma_{3,3}^z = 1$, and this deflection is only weakly reduced by the subsequent Gaussian energy filter (cf. Fig. 1). From a different viewpoint, the situation may also be seen as a small non-equilibrium system in contact with a large thermal bath (red and black vertices, respectively, in the sketch).

Accordingly, an obvious choice for the considered observable is the correlation between the initially disequilibrated sites, $A = \sigma_{2,2}^z \sigma_{3,3}^z$.

Incidentally, the ground-state energy of $H_0$ from (3) is approximately $-60$, whereas the infinite-temperature state has an energy of approximately $-1$. Hence, our choice of the target energy $E = -12$ should be reasonably generic and corresponds, as detailed in Supplementary Note 2.2, to an inverse temperature $\beta \approx 0.08$. Further examples for different target energies/temperatures can also be found in Supplementary Note 2.2.

In Fig. 1 we present numerical results, obtained by Suzuki-Trotter propagation, for the unperturbed system $H_0$ and for a sinusoidally driven system (1) with

$$f(t) = f_0\,\sin(2\pi t/T)\,,\tag{6}$$

yielding the solid black and blue lines, respectively.

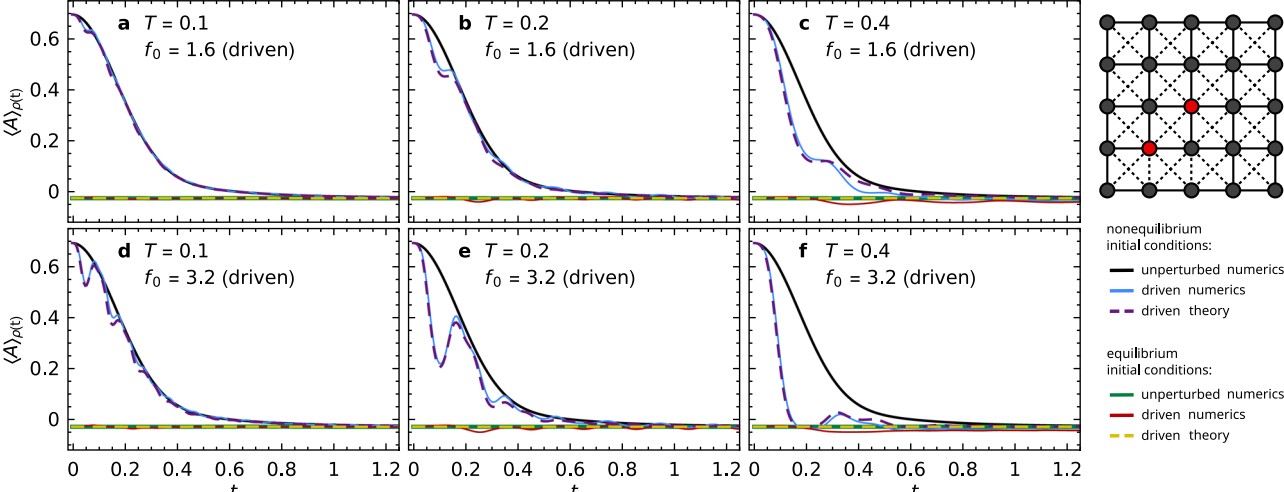

**Fig. 1 | Stalled response in a 5 × 5 lattice spin system.** Time-dependent expectation values $\langle A \rangle_{\rho(t)}$ of the magnetization correlation $A = \sigma_{2,2}^z \sigma_{3,3}^z$ are shown for a periodically driven system (see sketch) with Hamiltonian (1), (3), (4), (6). Solid black and blue lines: numerical results for non-equilibrium initial conditions (5) with $Q = \pi_{2,2}^+ \pi_{3,3}^+$, for driving amplitudes $f_0 = 0$ (unperturbed, black) and for driving

periods $T$ and amplitudes $f_0$ as indicated in each panel (driven, blue). Solid green and red lines: same but for equilibrium initial conditions (5) with $Q = \mathbb{1}$. Dashed lines: corresponding theoretical predictions (9), adopting the numerically obtained unperturbed behavior $\langle A \rangle_{\rho_0(t)}$, squared response function $|\gamma_r(t)|^2$ (by numerical integration of (10)), and thermal equilibrium value $A_{\mathrm{th}} = -0.026$ (see below Eq. (6)).

The key observation is that the driven (blue) and undriven (black) expectation values in Fig. 1 differ quite notably during the initial relaxation of the unperturbed system, but they become (nearly) indistinguishable upon approaching their (almost) steady long-time values. Moreover, both long-time values agree very well with the thermal expectation value $A_{th} \simeq -0.026$, obtained numerically by evaluating $A = \sigma_{2,2}^z \sigma_{3,3}^z$ in the microcanonical ensemble of the unperturbed system. In other words, the perturbations by the periodic driving get stalled upon thermalization of the undriven system.

To further highlight this phenomenon, let us also consider the analogous equilibrium initial conditions with $Q = 1$ in (5). Hence, the initial state populates the same energy window as in the non-equilibrium setting, but the observable expectation values now (approximately) assume the pertinent thermal equilibrium values[15–17]. The solid green and red lines in Fig. 1 illustrate the so-obtained numerical results for the unperturbed and the driven system. In particular, the initial expectation value is now very close to the thermal equilibrium value $A_{th} \simeq -0.026$. Moreover, the effects of the driving are indeed barely noticeable, and are even expected to become still smaller for larger system sizes, as detailed in Supplementary Note 2.3.

The bottom line of all these numerical findings is that the same system exhibits a quite significant response to the periodic driving away from thermal equilibrium, but hardly shows any reaction to the same driving as the unperturbed system approaches thermal equilibrium, or if it already started out near thermal equilibrium (stalled response).

Note that the driving amplitudes in Fig. 1 are far outside the linear response regime, as can be inferred, e.g., by comparing the blue curves of Fig. 1c and f (see also Supplementary Note 2.1). We also remark that for noncommuting perturbations and observables (as in Fig. 1), linear response theory generically excludes that there is no response at all. The main challenge is to understand why the non-linear response remains so weak at thermal equilibrium.

Likewise, the observable response becomes uninterestingly weak for extremely small or large driving periods $T$, regardless of the initial conditions and their proximity to thermal equilibrium. Hence, our focus here is on the natural regime of moderate $T$ values that are similar to, or slightly below the relaxation time of the unperturbed system, where the stalling effect is most pronounced and interesting. The interplay of the various time scales is further elaborated in Supplementary Note 1.1.

Finally, it is well-established that, for sufficiently large times, the driving will ultimately heat up the system towards a thermal steady state with infinite temperature[18–22]. However, it is equally well-established that this heating may often happen only very slowly, particularly for sufficiently small driving periods $T$[23–26]. Our present stalled response effect thus complements and substantially extends those previous predictions from Refs. [18–22].

## Theory

Our next goal is to establish an analytical theory for reasonably general classes of many-body quantum systems which explains these numerical findings. We start by collecting the basic ingredients and assumptions, then present the main result, and finally sketch the derivation.

First, we focus on initial states $\rho(0)$ with a well-defined macroscopic energy. Denoting by $E_\mu$ and $|\mu\rangle$ the eigenvalues and -vectors of the unperturbed Hamiltonian $H_0$, this means that non-negligible level populations $\langle\mu|\rho(0)|\mu\rangle$ only occur for energies $E_\mu$ within a sufficiently small energy interval $\Delta$, such that the density of states can be approximated by a constant $D_0$ throughout $\Delta$.

Second, within this energy interval $\Delta$, the matrix elements $V_{\mu\nu} := \langle\mu|V|\nu\rangle$ of the perturbation operator $V$ are assumed to exhibit a well-defined perturbation profile

$$\tilde{v}(E) := \left[|V_{\mu\nu}|^2\right]_E, \tag{7}$$

where $[\cdots]_E$ denotes a local average over matrix elements with $|E_\mu - E_\nu| \approx E$. The perturbation profile's Fourier transform is denoted as

$$v(t) := \int dE\, D_0\, \tilde{v}(E)\, e^{iEt}. \tag{8}$$

In passing, we note that at sufficiently high temperatures, $v(t)$ can be approximated by the two-point correlation function $\langle V(t)V\rangle_{\rho_{mc}}/2$, where $V(t) := e^{iH_0 t} V e^{-iH_0 t}$ and $\rho_{mc}$ is the microcanonical ensemble corresponding to the energy interval $\Delta$; see Supplementary Note 3 for details.

Third, the time-dependent perturbations $f(t)V$ in (1) should not become overly strong compared to $H_0$, so that establishing a connection between the unperturbed and driven systems remains sensible and the above mentioned heating effects stay reasonably weak.

In terms of the above introduced quantities, our main analytical result is the prediction

$$\langle A\rangle_{\rho(t)} = A_{th} + |\gamma_t(t)|^2 \left[\langle A\rangle_{\rho_0(t)} - A_{th}\right], \tag{9}$$

where $A_{th} = tr(\rho_{mc}A)$ is the thermal expectation value introduced below Eq. (6). The driving effects are encoded in the response function $\gamma_\tau(t)$, evaluated at $\tau = t$ in (9), which is obtained as the solution of the parametrically $\tau$-dependent family of integro-differential equations

$$\dot{\gamma}_\tau(t) = \int_0^t ds\, \gamma_\tau(s)\, \gamma_\tau(t-s)\, [a_\tau v(s) + b_\tau \ddot{v}(s)] \tag{10}$$

with initial condition $\gamma_\tau(0) = 1$ and coefficients

$$a_\tau := -[F_1(\tau)/\tau]^2, \quad b_\tau := [F_2(\tau)/\tau - F_1(\tau)/2]^2, \tag{11}$$

where $F_1(\tau) := \int_0^\tau dt\, f(t)$ and $F_2(\tau) := \int_0^\tau dt\, F_1(t)$. We emphasize that the theory and Eq. (10) in particular are nonlinear, which – in light of the numerically observed response characteristics (see Fig. 1) – is essential to faithfully reproduce the observed behavior.

To derive these results, we combined and advanced three major theoretical methodologies: (i) a Magnus expansion[27] for the propagator $\mathcal{U}(t)$ (see below Eq. (2)); (ii) a mapping of the time-dependent problem (1) to a parametrically $\tau$-dependent family of time-independent auxiliary systems; (iii) a typicality (or random matrix) framework[28–30] to determine the generic behavior (9) for the vast majority of all systems sharing the same $H_0$, $\tilde{v}(E)$, and $f(t)$. Details of the derivation are collected in the Methods.

Of the adopted techniques, the Magnus expansion in particular implies that such an approach only covers the transient dynamics up to a certain maximal time, which increases as the driving period $T$ becomes smaller. Since this maximal time has been related to the onset of heating[18,22,31], the result (9) does not capture such heating effects anymore. Yet it may well remain valid over a quite extended time interval since heating is suppressed exponentially for small $T$[7,8,23–26], see also Supplementary Note 1 for a more detailed discussion of the relevant time scales and of the response function $\gamma_\tau(t)$.

Due to the employed typicality framework, in turn, the prediction (9) may not reproduce the dynamics accurately in certain setups with strong correlations between the observable $A$ and the perturbation $V$. A more in-depth discussion of the expected regime of applicability is provided in the Methods.

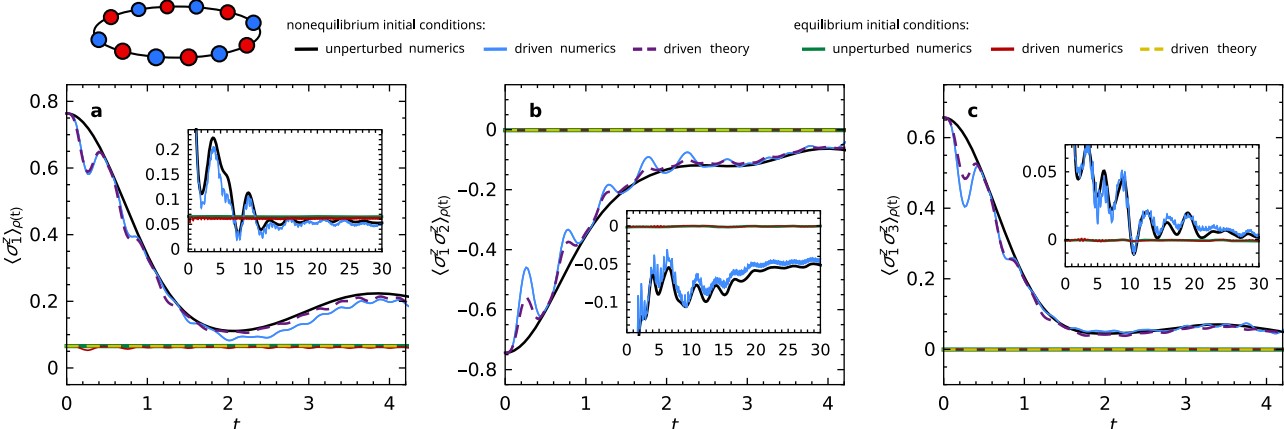

**Fig. 2 | Stalled response in a one-dimensional Ising-type model.** The unperturbed Hamiltonian $H_0$ in (1) is the transverse-field Ising model (see sketch), exhibiting periodic boundary conditions and additional next-nearest-neighbor couplings to break integrability, $H_0 := -J\sum_{j=1}^{L}(\sigma_j^x \sigma_{j+1}^x + \epsilon\,\sigma_j^x \sigma_{j+2}^x + g\,\sigma_j^z)$ with $J = \epsilon = g = \frac{1}{2}$ and $L = 24$. The driving operator is a longitudinal magnetic field, $V := -J\sum_{j=1}^{L}\sigma_j^x$. Time-dependent expectation values $\langle A\rangle_{\rho(t)}$ of (**a**) the single-site magnetization $A = \sigma_1^z$, (**b**) the nearest-neighbor correlation $A = \sigma_1^z\sigma_2^z$, and (**c**) the next-nearest-neighbor correlation $A = \sigma_1^z\sigma_3^z$ are shown for the periodically driven system (1), (6), with driving amplitude $f_0 = 4$ and period $T = 0.5$. Solid black and blue lines: numerical results for nonequilibrium initial conditions (5) with $|\phi\rangle = |\uparrow\downarrow\uparrow\downarrow\cdots\rangle$ (Néel state, see sketch), $Q = 1$, $E = -2.4$, and $\Delta E = 1$. (The corresponding inverse temperature, ground-state energy, and infinite-temperature energy are now approximately 0.2, $-18.5$, and 0, respectively, see also above Eq. (6).) Solid green and red lines: same but for equilibrium initial conditions (5), i.e., with a Haar-random state $|\phi\rangle$. Dashed lines: corresponding theoretical predictions (9), adopting the numerically obtained unperturbed behavior $\langle A\rangle_{\rho_0(t)}$, squared response function $|\gamma_t(t)|^2$ (by numerical integration of (10)), and thermal equilibrium values $A_{\text{th}} \simeq 0.066$, 0, 0 in (**a**–**c**), respectively. Insets: Same numerical data, but with rescaled $x$ and $y$ axes to display the long-time behavior.

## Interpretation and further examples

For a quantitative comparison of our theoretical prediction (9) to specific examples, some approximate knowledge of the perturbation profile (7) is clearly indispensable. Qualitatively, however, the theory quite remarkably allows us to make some largely general predictions without any such specific knowledge.

The first and foremost of these predictions is based on the general upper bound $|\gamma_t(t)| \leq 1$, whose detailed analytical derivation is provided in Supplementary Note 7 (see also Supplementary Note 1.2). It then immediately follows from (9) that the driving effects are strongly suppressed whenever $\langle A\rangle_{\rho_0(t)} \simeq A_{\text{th}}$, i.e., whenever the unperturbed system is close to thermal equilibrium. The latter in turn is true for all times $t$ if the unperturbed system is at thermal equilibrium from the outset, and for all sufficiently late times $t$ if the unperturbed system starts out far from equilibrium and is know to thermalize in the long run. Altogether, our stalled response phenomenon is thus analytically predicted to occur under very general circumstances.

Next we turn to a more detailed quantitative comparison of the theoretical prediction (9) with concrete numerical examples. For the setup considered in Fig. 1, exact diagonalization of a smaller system with $L = 4$[32] suggests that the perturbation profile $\tilde{v}(E)$ from (7) can be approximated very well by an exponential decay $\tilde{v}(0)\,e^{-|E|/\Delta_v}$. Utilizing Ref. 30, one moreover finds for the $L = 5$ system in the relevant energy window the numerical estimates $\tilde{v}(0)D_0 \simeq 3.6$ and $\Delta_v \simeq 3.0$, yielding $v(t)$ via (8). All quantities entering the theoretical prediction (9)–(10) are thus either numerically available [$\langle A\rangle_{\rho_0(t)}$, $A_{\text{th}}$] or otherwise known [$v(t)$, $a_\tau$, $b_\tau$], i.e., there remains no free fit parameter.

As can be inferred from the solid blue and dashed purple lines in Fig. 1, the theory indeed describes the nontrivial details of the driven dynamics remarkably well. Notably, it reproduces the pronounced drop compared to the unperturbed curve around $t = T/2$ and the quite surprising comeback around $t = T$. Moreover, it indeed also explains the stalled response behavior in Fig. 1 very well, for initial conditions both close to and far from thermal equilibrium.

Within the framework of Floquet theory, a related, but distinct effect is well-known under the name "Floquet prethermalization"[7,8,20,23,24,26,33,34]: The dynamics described by the Floquet Hamiltonian approaches a prethermal plateau value before

heating becomes significant and pushes the system towards infinite temperature. However, the dynamics encoded in the Floquet Hamiltonian only agrees with the actual dynamics of the driven system stroboscopically, i.e., only at integer multiples of the driving period. A prethermal plateau of the Floquet-Hamiltonian dynamics therefore still leaves room for strong oscillations of the actual dynamics between the stroboscopic time points where both agree. Accordingly, the salient new insight provided by our present results is that no such strong oscillations are observed if the unperturbed system relaxes to or starts out from a thermal equilibrium state. In other words, our stalled response effect amounts to a highly nontrivial extension of the established Floquet prethermalization phenomenon since it means that the plateau value is assumed not only stroboscopically, but even continuously in $t$. An extended discussion of the relation between our approach and Floquet theory can be found in Supplementary Note 4.

As a second example, we consider a nonintegrable variant of the transverse-field Ising model in Fig. 2, see the figure caption for details. We particularly emphasize that, for variety and in contrast to Fig. 1, this setup consists of a one-dimensional system and globally out-of-equilibrium initial conditions.

Qualitatively, the numerical results in Fig. 2 once again confirm the main message of our paper, namely the occurrence of stalled response: Initially, the dynamics shows a pronounced response when starting away from equilibrium (solid black vs. blue lines). Stalling of that response appears as the unperturbed system approaches thermal equilibrium, meaning that the oscillations caused by the driving become smaller and smaller. This is highlighted in the insets, in particular. (A special feature of this example is that already the unperturbed system (black lines) exhibits a relatively complex and long-lasting relaxation process.) Likewise, the effects of the driving are barely visible on the scale of the plot when starting directly from a thermal equilibrium state (solid green vs. red lines).

For a quantitative comparison of the numerical results with the theoretical prediction (9), we assume, as in the previous example, an approximately exponential perturbation profile $\tilde{v}(E) = \tilde{v}(0)e^{-|E|/\Delta_v}$ [cf. Eq. (7)], and use again the theory from Ref. 30 to estimate $\tilde{v}(0)D_0 \simeq 0.46$ and $\Delta_v \simeq 0.6$. The resulting theoretical curves in Fig. 2 (dashed lines) describe the numerics reasonably well in the initial regime. In

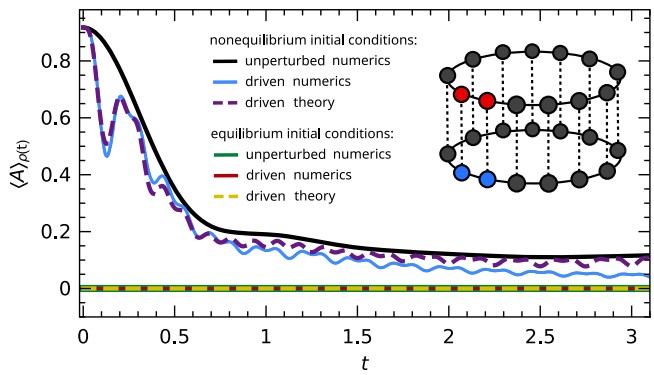

**Fig. 3 | Imperfect stalling upon breaking a conservation law.** Time-dependent expectation values $\langle A \rangle_{\rho(t)}$ of the single-site magnetization $A = \sigma_{1,1}^z$ are shown for a periodically driven $2 \times 14$ spin double-chain (see inset) with Hamiltonian (1), (6), (12), (13), and driving period $T = 0.25$. Solid black and blue lines: numerical results for non-equilibrium initial conditions (5) with $Q = \pi_{1,1}^+ \pi_{1,2}^+ \pi_{2,1}^- \pi_{2,2}^-$, for driving amplitudes $f_0 = 0$ (unperturbed, black) and $f_0 = 3.2$ (driven, blue). Solid green and red lines: same but for equilibrium initial conditions (5) with $Q = \mathbb{1}$. Dashed lines: corresponding theoretical predictions (9), obtained as in Fig. 1 but with $A_{\mathrm{th}} = 0$.

accordance with the discussion below Eq. (11), for larger times the theory is no longer quantitatively very accurate (but still correctly predicts the occurrence of stalling per se). For this reason, no dashed lines are shown in the insets.

Yet another interesting general prediction of the theory (9) (see also beginning of this section) is that noticeable effects of the driving (as encoded in $|\gamma_t(t)|^2$) may actually persist even beyond the relaxation time scale of the unperturbed system if its long-time expectation value $\bar{A}_0 := \overline{\langle A \rangle_{\rho_0(t)}}$ (infinite time average) differs from the thermal value $A_{\mathrm{th}}$. This can happen, for example, if the perturbation $V$ breaks a conservation law of $H_0$.

To verify this prediction, we consider a third example in Fig. 3. Here the unperturbed system consists of two isolated spin chains of $L = 14$ sites with periodic boundary conditions and Hamiltonian

$$H_0 := H^{(1)} + H^{(2)}, \quad H^{(i)} := \sum_{j=1}^{L} \boldsymbol{\sigma}_{i,j} \cdot \boldsymbol{\sigma}_{i,j+1}, \tag{12}$$

while the perturbation in (1) connects the chains sitewise,

$$V := \sum_{j=1}^{L} \boldsymbol{\sigma}_{1,j} \cdot \boldsymbol{\sigma}_{2,j}; \tag{13}$$

see also the sketch in the inset. The initial state is again of the form (5) with $E = -14$ and $\Delta E = 4$, restricted to the sector with vanishing $S^z := \sum_j (\sigma_{1,j}^z + \sigma_{2,j}^z)$. (The corresponding inverse temperature, ground state energy, and infinite-temperature energy are now approximately $0.12, -50$, and $-1$, respectively see also above Eq. (6).) However, for the nonequilibrium setup we now fix two spins in the "up" state for the first chain and two in the "down" state for the second chain (red and blue, respectively, in the sketch), i.e., $Q := \pi_{1,1}^+ \pi_{1,2}^+ \pi_{2,1}^- \pi_{2,2}^-$. Since the two chains $(i = 1, 2)$ do not interact in the unperturbed system, their magnetizations $S_i^z := \sum_j \sigma_{i,j}^z$ are conserved individually, and thus maintain their initial expectation values $2$ and $-2$, respectively, under evolution with $H_0$. In the driven system, by contrast, only the total $S^z := S_1^z + S_2^z$ is conserved. Choosing the single-site magnetization $A = \sigma_{1,1}^z$ as our observable, we thus find by symmetry that $\bar{A}_0 = 2/L$ is the long-time expectation value of the unperturbed dynamics, whereas the thermal value of the joint system is $A_{\mathrm{th}} = 0$.

The numerics in Fig. 3 (solid blue line) visualizes the aforementioned imperfect stalling upon breaking a conservation law: The

suppression of the response is the stronger the closer the unperturbed system is to thermal equilibrium. Crucially, however, the driving effects still remain visible even when the unperturbed dynamics has essentially reached its nonthermal long-time value $\bar{A}_0$. Altogether, this confirms the prediction of (9) that proximity to thermal equilibrium is indeed the decisive condition for stalled response and not, for example, relaxation of the unperturbed system. Furthermore, this example highlights once again that stalled response and Floquet prethermalization are distinct effects: The present system exhibits Floquet prethermalization, meaning that the stroboscopic dynamics approaches a stationary plateau, but no stalled response since $\langle A \rangle_{\rho(t)}$ continues to oscillate.

For a quantitative comparison with the theory (9), we again adopt the same ansatz as before and estimate $\bar{v}(0) D_0 = 0.98$ and $\Delta_v = 4.2$ via[30]. The so-obtained prediction (9) (dashed purple) agrees rather well with the numerics for $t \lesssim 1$. At later times, the quantitative deviations between the prediction and the numerics increase. As suggested below (11) and discussed in more detail in the Methods, we can attribute these deviations to the adopted Magnus expansion and its truncation at second order. Yet the above mentioned general qualitative prediction of our theory remains valid nonetheless.

### Basic physical mechanisms

Intuitively, the basic physics behind all our above mentioned numerical and analytical findings can also be understood by means of the following simple arguments: As long as heating is insignificant, we may focus on the dynamics within the initially populated energy interval $\Delta$ (see above (7)). Denoting by $P$ the projector onto the eigenstates $|\mu\rangle$ with $E_\mu \in \Delta$, the Hamiltonian $H(t)$ from (1) can thus be reasonably well approximated by its projection/restriction $\tilde{H}(t) := PH(t)P$ to $\Delta$. Since the microcanonical ensemble $\rho_{\mathrm{mc}} := P/\mathrm{tr}\{P\}$ commutes with $\tilde{H}(t)$, it is a stationary state with respect to $\tilde{H}(t)$. Within the present approximation, a system in thermal equilibrium is thus completely unaffected by the periodic driving, and analogously the effects remain weak if the system is in a state close to thermal equilibrium. (Incidentally, the relaxation of a non-equilibrium initial state under $\tilde{H}(t)$ can be heuristically understood by similar arguments as in Ref. [19].) On the other hand, subleading effects like small remnant oscillations and slow heating cannot be understood within this simplified picture. Rather, these effects must be attributed to the neglected corrections $H(t) - \tilde{H}(t)$ and, as a consequence, are intimately connected with each other.

A complementary, and even more simplistic argument is based on the well-established fact[35–37] that the vast majority of all pure states with energies in $\Delta$ behave akin to $\rho_{\mathrm{mc}}$ for sufficiently large many-body systems. This so-called typicality property suggests that once the system has reached (or starts out from) such a state, it remains within this vast majority in the absence as well as in the presence of the periodic driving.

Essentially, our stalled response effect thus seems to be the result of a subtle interplay between the system's many-body character and intriguing peculiarities of thermal equilibrium states. The above intuitive arguments moreover suggest that the indispensable prerequisites for stalled response per se may be substantially weaker than those of our analytical theory (see also Supplementary Note 2.4).

## Discussion

Our core message is that the same many-body system may either exhibit a quite significant response when perturbed by a periodic driving, or may not show any notable reaction to the same driving, depending on whether the unperturbed reference system finds itself far from or close to thermal equilibrium. We demonstrated this stalled response effect by numerical examples, and further substantiated it by sophisticated analytical methods and by simple physical arguments.

Previous theoretical and experimental studies of periodically driven many-body systems (e.g., Refs. 6–8,13,19–21,23,24,31,33,34,38,39

among many others) have been very successful in characterizing the long-term properties of such systems, including heating effects[18–22,31,40,41] and their suppression[6–8,21,23–26,38,42]. The latter, in particular, facilitates the phenomenon of Floquet prethermalization[7,8,20,23,24,26,33,34], a long-lived, stroboscopically quasistationary phase which has been exploited, for instance, to design various meta materials with promising topological and dynamical properties[11–14,34].

Complementary to those long-term features for discrete time points, our present focus is on how a many-body system approaches such prethermal regimes continuously in time. Overall, we thus arrive at the following general picture for periodically driven systems with moderate driving periods and amplitudes: Given a thermalizing unperturbed system that is prepared sufficiently far from equilibrium, the periodic perturbations generically lead to quite notable response effects on short-to-intermediate time scales. Subsequently, the expectation values approach a (nearly) time-independent behavior. On even much larger time scales, the system finally heats up to infinite temperature, manifesting itself in a slow drift of the expectation values towards their genuine infinite-time limits.

In principle, our predictions can be readily tested with presently available techniques in, for example, cold-atom[1–6] or polarization-echo[6–8] experiments. In practice, previous experimental (as well as theoretical) investigations mostly focused on the long-time behavior and stroboscopic dynamics. A notable exception is the NMR experiment from Ref. 8: In Figs. 3(a) and 5(a,b) therein, the NMR signal of the initially out-of-equilibrium system undergoes vigorous oscillations at first (called "transient approach" in Ref. 8). Then, their amplitude gradually decreases as the running mean approaches a quasistationary value (called "prethermal plateau"[8]). Even later, the only noticeable effect of the driving is a slow drift as the system heats up (called "unconstrained thermalization"[8]). Unfortunately, the available experimental details are not sufficient to compare the measurements quantitatively with our analytical theory (9). Nevertheless, the observed NMR signal clearly shows the general qualitative features of stalled response as predicted by Eq. (9).

## Methods

We first lay out the three main steps in the derivation of (9)–(10), and subsequently address the expected validity regime of the employed approximations.

### Magnus expansion

The time evolution of the driven quantum system with Hamiltonian $H(t)$ from (1) is encoded in the propagator $\mathcal{U}(t)$ introduced below Eq. (1), which satisfies the Schrödinger-type equation $\frac{d}{dt}\mathcal{U}(t) = -iH(t)\mathcal{U}(t)$. Whereas this equation is formally solved by an (operator-valued) exponential for time-independent Hamiltonians, no such simple solution is available for the driven case. To make progress while keeping the setting as general as possible, we adopt a Magnus expansion[27] of the propagator, writing

$$\mathcal{U}(t) = e^{\Omega(t)}, \quad \Omega(t) = \sum_{k=1}^{\infty} \Omega_k(t), \tag{14}$$

where the individual terms $\Omega_k(t)$ in the exponent consist of integrals over $k-1$ nested commutators of $H(t)$ at different time points. The virtue of the Magnus series compared to other expansion schemes (e.g., a Dyson series) is that $\mathcal{U}(t)$ remains unitary when truncating (14) at a finite order.

For Hamiltonians of the specific form (1), the first two terms of the general Magnus expansion (see, e.g., Ref. 27) can be readily rewritten as

$$\Omega_1(t) = -i\left[H_0 t + F_1(t)V\right], \tag{15a}$$

$$\Omega_2(t) = \left[F_2(t) - \frac{t}{2}F_1(t)\right][V, H_0], \tag{15b}$$

where $[V, H_0] := VH_0 - H_0 V$ (commutator), and $F_{1,2}(t)$ are defined below Eq. (11).

### Mapping to auxiliary systems

Adopting the Magnus expansion (14), the propagator $\mathcal{U}(t) = e^{\Omega(t)}$ assumes an exponential form similar to the case of time-independent Hamiltonians. However, the time dependence of the exponent is generally still complicated. To proceed, we introduce a one-parameter family of time-independent auxiliary Hamiltonians

$$H^{(\tau)} := i\Omega(\tau)/\tau, \tag{16}$$

where $\tau > 0$ is treated as an arbitrary but fixed parameter. Starting from the same initial state $\rho(0)$ as in the actual system of interest, any of these Hamiltonians $H^{(\tau)}$ generates a time evolution with the state at time $t$ given by

$$\rho(t, \tau) := e^{-iH^{(\tau)}t}\rho(0)e^{iH^{(\tau)}t}. \tag{17}$$

Since $\rho(t) = \mathcal{U}(t)\rho(0)\mathcal{U}(t)^\dagger$, the combination of Eqs. (14), (16), and (17) implies that the state $\rho(t)$ of the driven system of interest coincides with the time-evolved state of the auxiliary system $H^{(t)}$ at time $t$, i.e.,

$$\rho(t) = \rho(t, t). \tag{18}$$

Hence finding the dynamics of the original driven system is equivalent to determining the behavior of all the auxiliary systems with time-independent Hamiltonians $H^{(\tau)}$ up to time $t = \tau$, respectively.

Restricting ourselves to the second order of the Magnus expansion, we adopt Eqs. (15) in (16) to approximate the auxiliary Hamiltonians as

$$H^{(\tau)} \simeq H_0 + V^{(\tau)} \tag{19}$$

with

$$V^{(\tau)} := \frac{F_1(\tau)}{\tau}V + \left[\frac{F_2(\tau)}{\tau} - \frac{F_1(\tau)}{2}\right]i[V, H_0], \tag{20}$$

thereby splitting off the $\tau$-independent reference Hamiltonian $H_0$.

### Typicality framework

It is empirically well established that the macroscopically observable behavior of systems with many degrees of freedom can be described by a few effective characteristics despite the vastly complicated dynamics of their individual microscopic constituents. Detecting and separating the macroscopically relevant properties of a many-body system from the intractable microscopic details can arguably be considered as the paradigm of statistical mechanics. The final component of our toolbox to describe the driven many-body dynamics aims at adopting such an approach to the observable expectation values $\langle A \rangle_{\rho(t)}$.

To this end, we start with the Hamiltonian $H(t) = H_0 + f(t)V$ from (1) and temporarily consider an entire class (or a so-called ensemble) of similar driving operators $V$. Ideally, we would like to establish that all members of such an ensemble exhibit the same observable dynamics. In practice, what is analytically feasible is a slightly weaker variant of such a statement. Namely, we demonstrate that nearly all members $V$

of the ensemble show in very good approximation the same *typical behavior*, and that the fraction of exceptional members, leading to noticeable deviations from the typical behavior, is exponentially small in the system's degrees of freedom.

In essence, the defining characteristic of the considered ensembles is the perturbation profile $\tilde{v}(E)$ from (7). Introducing the symbol $\mathbb{E}[\cdots]$ to denote the average over the $V$ ensemble, the matrix elements $V_{\mu\nu}$ are treated as independent (apart from the Hermiticity constraint, $V_{\mu\nu} = V^*_{\nu\mu}$) and unbiased ($\mathbb{E}[V_{\mu\nu}] = 0$) random variables with variance $\mathbb{E}[|V_{\mu\nu}|^2] = \tilde{v}(E_\mu - E_\nu)$. Hence the property (7) of the true perturbation is built into the ensemble in an ergodic sense, i.e., upon replacing local averages $[\cdots]_E$ (see below Eq. (7)) by ensemble averages $\mathbb{E}[\cdots]$. Due to a generalized central limit theorem (cf. Supplementary Note 6), these first two moments are essentially the only relevant characteristics of the $V$ ensemble, i.e., the precise distribution of the $V_{\mu\nu}$ can still take rather general forms. A detailed definition of the admitted ensembles is provided in Supplementary Note 5.

For time-independent Hamiltonians of the form $H = H_0 + \lambda V$ with a constant (time-independent) perturbation, it was demonstrated in Refs. [29,30] that those ensembles can indeed be employed to predict the observed dynamics in a large variety of settings. In the following, we will extend the underlying approach to the auxiliary Hamiltonians $H^{(\tau)}$ of the form (19). The distribution of the $V_{\mu\nu}$ thus induces a distribution of the matrix elements $V^{(\tau)}_{\mu\nu} := \langle\mu|V^{(\tau)}|\nu\rangle$ of $V^{(\tau)}$ from (20). In particular, we obtain $\mathbb{E}[V^{(\tau)}_{\mu\nu}] = 0$ and, together with the definitions (7), (11), and (20),

$$\mathbb{E}\left[|V^{(\tau)}_{\mu\nu}|^2\right] = -\left[a_\tau + \left(E_\mu - E_\nu\right)^2 b_\tau\right]\tilde{v}(E_\mu - E_\nu). \tag{21}$$

As a first step of our typicality argument, we then calculate the ensemble average $\mathbb{E}[\langle A\rangle_{\rho(t,\tau)}]$ of the time-evolved expectation values. Deferring the details to Supplementary Note 6, we eventually obtain the relation

$$\mathbb{E}[\langle A\rangle_{\rho(t,\tau)}] = A_{\text{th}} + |\gamma_\tau(t)|^2\left[\langle A\rangle_{\rho_0(t)} - A_{\text{th}}\right]. \tag{22}$$

Here a Fourier transformation relates the response function (see above (10)) via

$$\gamma_\tau(t) = \frac{1}{\pi}\lim_{\eta\to 0^+}\int dE\, e^{iEt}\,\text{Im}\,G(E - i\eta, \tau) \tag{23}$$

to the function $G(z, \tau)$, which solves

$$G(z,\tau)\left[z + \int dE\, D_0\, G(z - E, \tau)\left(a_\tau - E^2 b_\tau\right)\tilde{v}(E)\right] = 1 \tag{24}$$

and encodes the ensemble-averaged resolvent of $H^{(\tau)}$ via $\mathbb{E}[(z - H^{(\tau)})^{-1}] = G(z - H_0, \tau)$. In Supplementary Note 7, we furthermore show that Eqs. (23) and (24) imply the relation (10) for $\gamma_\tau(t)$.

As a next step, we turn to the deviations $\xi(t,\tau) := \langle A\rangle_{\rho(t,\tau)} - \mathbb{E}[\langle A\rangle_{\rho(t,\tau)}]$ between the driven dynamics induced by one particular perturbation operator $V$ and the average behavior. More explicitly, we inspect the probability $\mathbb{P}(|\xi(t,\tau)| \geq x)$ that a randomly selected perturbation $V$ generates deviations $\xi(t,\tau)$ that are larger than some threshold $x$. As explained in more detail in Supplementary Note 8, we can find a constant $\delta = 10^{-\mathcal{O}(N_{\text{dof}})}$ (decreasing exponentially with the system's degrees of freedom $N_{\text{dof}}$) such that

$$\mathbb{P}(|\xi(t,\tau)| \geq \delta\Delta_A) \leq \delta, \tag{25}$$

where $\Delta_A$ is the measurement range of $A$ (difference between its largest and smallest eigenvalues). In other words, observing deviations which exceed some exponentially small threshold value becomes exponentially unlikely as the system size increases, a phenomenon that is also sometimes called "concentration of measure" or "ergodicity" in the literature. Consequently,

$$\langle A\rangle_{\rho(t,\tau)} \simeq \mathbb{E}[\langle A\rangle_{\rho(t,\tau)}] \tag{26}$$

becomes an excellent approximation for the vast majority of perturbations $V$ in sufficiently large systems. Combining Eqs. (18), (22), and (26), we thus finally recover our main result (9).

## Limits of applicability

The class of systems whose Hamiltonian can be written in the form (1) is extremely general. However, the methods described above contain three major assumptions or idealizations that restrict the types of admissible setups to some extent.

The first issue arises when adopting the Magnus expansion (14) for the propagator $\mathcal{U}(t)$. The question of its convergence is generally a subtle issue and rigorously guaranteed in full generality only up to times $t$ such that the operator norm $\|H(s)\|$ satisfies $\int_0^t ds\,\|H(s)\| < \pi$, but can extend to considerably longer times in practice nonetheless[27]. Due to the extensive growth of $H(t)$ with the degrees of freedom, guaranteed convergence is thus very limited for typical many-body systems, but the expansion can still remain valuable as an asymptotic series for short-to-intermediate times[23,33]. For periodically driven systems in particular, the (Floquet-)Magnus series amounts to a high-frequency expansion and thus works best for small driving periods $T$[27,43]. More generally, the smaller the characteristic time scale of the driving protocol $f(t)$ is, the larger is the time up to which the expansion offers a satisfactory approximation at any fixed order.

Physically, the breakdown of the Magnus expansion has been related to the onset of heating[18,22,31]. Generically, many-body systems subject to perpetual driving are expected to absorb energy indefinitely and heat up to a state of infinite temperature[18–22], unless there are mechanisms preventing thermalization such as an extensive number of conserved quantities[38,42] or many-body localization[21,39,44]. Nevertheless, under physically reasonable assumptions about the system, such as locality of interactions, it has been shown that the heating rate is exponentially small in the driving frequency[23–26]. For sufficiently fast driving, therefore, energy absorption is essentially suppressed for a long time and the Magnus expansion can provide a good description of the dynamics. A more quantitative discussion of the interdependence of the relevant time scales is provided in Supplementary Note 1.1.

In summary, the Magnus expansion is expected to work as long as the state $\rho(t)$ stays roughly within the initially occupied microcanonical energy window $\Delta$ of the unperturbed reference Hamiltonian introduced above Eq. (7). Consequently, the stalled-response effect and the applicability of the prediction (9) are generally expected to persist for longer times at larger initial temperatures because the relative influence of heating is smaller in this case. Furthermore, higher temperatures come with a higher density of states, such that finite-size effects are smaller, too. The temperature dependence is discussed in more detail in Supplementary Note 2.2.

A second limitation is our truncation of the Magnus expansion at second order. In general, this will further restrict applicability towards shorter times and/or faster driving, but still leaves room for a broad and interesting parameter regime as demonstrated examplarily in Figs. 1–3. In principle, including higher-order terms may be possible, even though it leads to severe technical complications in the typicality calculation outlined above (see also Supplementary Note 6), and is thus beyond the scope of our present work. Besides the response function $\gamma_t(t)$, higher-order corrections are also expected to affect the long-time value ($A_{\text{th}}$ in Eq. (9)): It is well known from Floquet theory that this plateau value of Floquet prethermalization is controlled by the Floquet Hamiltonian[23,24,26,33,34]. The latter agrees with $H_0$ to lowest

order, but can yield different long-time behavior in general, even though the corrections are generically expected to be small[23].

A third potentially limiting factor for the applicability of our present approach is the typicality framework, within which we introduce ensembles of matrix representations $V_{\mu\nu}$ of the driving operator $V$ in the eigenbasis of the reference Hamiltonian $H_0$. Our main result states that the observable dynamics of nearly all members $V$ of such an ensemble is described by Eqs. (9) and (10) (up to the limitations discussed earlier). The final point to establish is that the true (non-random) driving operator $V$ of actual interest is one of those typical members of the ensemble, which evidently requires a faithful modeling of the system's most essential properties with regard to the observable dynamics.

The classes of perturbation ensembles considered here are a compromise between what is physically desirable and mathematically feasible. From a physical point of view, we would like to emulate the matrix structure of realistic models as closely as possible. We therefore explicitly incorporate the possibility for sparse (most $V_{\mu\nu}$ are strictly zero) and banded (the typical magnitude $|V_{\mu\nu}|$ decays with the energy separation $|E_\mu - E_\nu|$ of the coupled levels) perturbation matrices. These features indeed commonly arise as a consequence of the local and few-body character of interactions in realistic systems as supported by semiclassical arguments[45,46], analytical studies of lattice systems[47,48], and a large number of numerical examples (e.g. Refs. 49–51). Similar assumptions are also well-established in random matrix theory and in the context of the eigenstate thermalization hypothesis[28,52–54]. On the other hand, the geometry of the underlying model and the structure of interactions (for instance their locality) are not explicitly taken into account. Therefore, the existence of macroscopic transport currents as a consequence of macroscopic spatial inhomogeneities can likely invalidate the prediction (9)–(10), at least for observables $A$ which are sensitive to such initial spatial imbalances and their equalization in the course of time.

This is ultimately related to our idealization of statistically independent matrix elements $V_{\mu\nu}$ for $\mu \leq \nu$. In any realistic system, some of the matrix elements will certainly mutually depend on each other. However, it is generally hard to identify (let alone quantify) potential correlations in any given system, so independence may also be understood as unbiasedness in the absence of more detailed information. Moreover, mild correlations will often not have a noticeable impact on the properties relevant for the observable dynamics[55].

A specific case where correlations can become relevant, though, are observables $A$ that are strongly correlated with the perturbation $V$, most notably if $A = V$. Since we keep the observable fixed when calculating ensemble averages, most members of the $V$ ensemble will obviously violate such a special relationship. Unfortunately, it is not straightforwardly possible to adapt the method such that the case $A = V$ can be described as well because including $A = V$ in the ensemble averages would also affect the unperturbed reference dynamics $\langle A \rangle_{\rho_0(t)}$. Numerical explorations and further discussions of this case are provided in Supplementary Note 2.4. Notably, the qualitative predictions of the theory (9) and, in particular, the occurrence of stalled response can still be seen for the observable $A = V$.

For the rest, we emphasize that it is not necessary for all members $V$ of a certain ensemble to be physically realistic. The decisive question is whether their majority embody the key mechanism underlying the observable dynamics in the same way as the true system of interest. To give an example from textbook statistical mechanics, a large part of states contained in the canonical ensemble (as a mixed density operator) will be unphysical, and yet its suitability to characterize macroscopically observable properties of closed systems in thermal equilibrium is unquestioned provided that the temperature as the pertinent macroscopic parameter is chosen appropriately.

More generally, the probabilistic nature of the result implies that any given system can show deviations even if all prerequisites are formally fulfilled, but the probability for such deviations is exponentially suppressed in the system's degrees of freedom, cf. Eq. (25). For generic many-body systems, we therefore cannot but conclude that Eqs. (9)–(10) are expected to hold unless there are specific reasons to the contrary. The explicit example systems from Figs. 1–3 only corroborate this observation, noticeably even though the number of degrees of freedom is still far from being truly macroscopic in those systems.

## Data availability
The data generated in this study are provided in the Supplementary Information/Source Data file. Source data are provided with this paper.

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

## Acknowledgements

This work was supported in parts by the Deutsche Forschungsgemeinschaft (DFG, German Research Foundation) within the Research Unit FOR 2692 under Grant No. 355031190 (PR). We acknowledge support for the publication costs by the Open Access Publication Fund of Bielefeld University and the Deutsche Forschungsgemeinschaft (DFG).

## Author contributions

This work was carried out by L.D. and P.R.

## Funding

## Competing interests

The authors declare no competing interests.
