## [Peer Review File · Nature Communications]

REVIEWER COMMENTS

Reviewer #1 (Remarks to the Author):

In this manuscript, Dabelow and Reimann obtain a result for the behaviour of local observables in periodically-driven systems, in particular, that the heating (or generally the time evolution) of such observables slows down when the undriven system is close to a thermal state.

This is an important result by itself, and the existence of an analytical derivation makes this even more so--such results are not frequently possible in such systems. They also demonstrate its validity by numerical experiments--but see below for some questions on that.

Overall, given the amount of activity on the subject of Floquet systems, it is clear that this is an important and timely topic, and their result is likely to serve as a foundation for future work.

This manuscript does a good job overall presenting their analytical result. Their numerical demonstrations thereof are convincing as far as they go but, I think, incomplete. I therefore have a few questions before I feel I can recommend publication:

1. The numerical examples are at the fairly high temperatures $\beta=0.08, 0.2$ and 0.12 ; for all of these the observable at equilibrium is close to its infinite-temperature value. Can the authors discuss and show results from states where the temperature is lower? Is this technically difficult because the density of states at low temperatures is small for accessible system sizes, for example, or for some other reason? The authors should explain better these aspects of their work--either by presenting results with initial states with E (in their Eq. 5) further away from the middle of the spectrum, or discussing why that is not possible (or why their results do not apply to that regime, if they do not). At the moment all I can find about this in the main text is that the theory is valid for longer times at higher energies. Surely they have a more nuanced picture given all the work that they did, so why not present it?

2. Do the authors have any results on the general behaviour of γ ? For instance, is it monotonic for some classes of $v(s)$, or anything like that? (I realise it's not always monotonic--I'm simply asking whether they can show anything at all about this).

I also have a few more minor comments:

1. Under Eq. 7, they say the average of the perturbation is over $\langle E_{\mu-E_{\nu}} \rangle \approx E$ and not $\langle |E_{\mu-E_{\nu}}| \rangle \approx E$. Is the distinction important? It seems to me that it is not. Is this correct?

2. Presentation-wise, the Methods sections do give useful information that can be pieced together to get a sense of how the main result of Eq. 9 would be derived, but the way they are presented leaves to the reader the task of putting them together. Instead of just presenting the pieces and letting the reader fit them together, perhaps a paragraph explaining how they fit could be added.

3. Possibly not that important, but chronologically the first paper to show MBL is stable under driving is PRL 115, 030402 (2015) (it and Ref 20 appear to conclude similar things almost contemporaneously, but this is slightly earlier). This isn't really relevant to the correctness of their conclusions--this is just a comment.

I am looking forward to the author's responses to my two comments on the substance of the manuscript.

Reviewer #2 (Remarks to the Author):

The paper develops a theoretical framework to describe the response of a quantum system, initially in an out of equilibrium state, subject to periodic driving. The main theoretical result is eq.(9) which predicts a particularly modest impact of driving, which can not significantly alter thermalization dynamics, as measured by some local observable. Specifically, since γ in eq. (9) can not be larger than 1, the authors emphasize that the driving can not preclude or delay thermalization. This result is potentially interesting, and somewhat surprising. It might depend on the choice of driving operator V and the monitored observable A . Could it be that the modest impact of driving is due to small value of the 2pt function between A and V ? I guess if $A=V$ or more generally $\langle A(t)V \rangle$ is large (here time evolution is by H_0), equation 9 with $|\gamma| \leq 1$ may not apply. I suggest authors investigate and comment on this point in the paper.

Another comment, to have predictive power, eq. (9) require function v from eq. (7) as an input. Can one define (7) through the Fourier of 2-point function of V , using arguments based on ETH? Without that, if one has to rely on direct diagonalization, the whole point of (9) disappears -- with the exact diagonalization one can calculate the dynamics of eq. (2) exactly. So, I suggest, if this is possibly to reformulate (9) solely in terms of observable quantities, e.g. $\langle V(t)V \rangle$ at temperature specified by ρ , etc.

An interesting situation suggested by the authors is when asymptotic value of $\langle A \rangle(t)$, subject to underdriven evolution, is different from thermal equilibrium. The authors considered such a case, but as I can judge from their Fig. 3, at late times their theoretical prediction (blue dashed line) is systematically different from the exact result (solid blue line). I suggest the authors comment on this point.

A few more comments about the manuscript. The authors called the modest response to driving "stalled" - I'm not sure why they choice this word and what it should convey. They also several times call their finding "astonishing." I believe they should explain in the manuscript what they default expectation was which warrants this adjective. Finally, they use blue lines (solid and dashed) in Fig.1, and this choice is very confusing, making the lines almost indistinguishable. Finally, in the beginning of the paper the authors mentioned recent NMR experiments as an example of systems where their theory applies. But toward the end they admitted that available information (about the experimental setting etc.) is not sufficient to make an accurate comparison, validating their theory. I therefore suggest to remove the mention of NMR from the introduction.

Reviewer #3 (Remarks to the Author):

This paper addresses the relaxation phenomena of quantum many-body systems under periodic driving fields. The manuscript is well organized.

The authors developed the theory of relaxation phenomena (Eqs. (9),(10),(11)) based on the Floquet-Magnus expansion up to the second order and argument with the typicality. The relaxation phenomena are governed mainly by the unperturbed Hamiltonian, and the values of initial relaxation go to the thermal equilibrium (equivalently to microcanonical) values given by the initial states. The periodic driving does not significantly contribute to the overall structure of the relaxation, which the authors dub stalled response. These observations are highly connected to the well-known Floquet prethermalization.

I could not understand how new these observations are at the moment.

(1) So far, people have (presumably) thought that the prethermalization plateau is given by the Gibbs state with the Floquet Hamiltonian computed from the Floquet Magnus expansion, where the initial state gives the effective temperature. In this paper, the initial relaxation goes to the value of the thermal state of the unperturbed Hamiltonian that does not contain additional Floquet orders. A new aspect is that the prethermal-relaxation is dominated only by the unperturbed Hamiltonian.

(2) The authors repeatedly say that the stalled response is astonishing. However, the standard picture of Floquet prethermalization is that the state relaxes onto the thermalized state of Floquet Hamiltonian. Based on this picture, insensitivity to the driving field, starting from the thermal state, is already expected.

(3) The sentence in page 4: 'Our present result amounts to a highly nontrivial extension of this previously established insight. Namely, it implies that the plateau value is not only approached stroboscopically but continuously'.

I could not understand why the continuous approach is surprising, which is something expected. If this observation is highly nontrivial, the authors should explain the detail.

(4) Question: With your nice formula Eq.(9), can you estimate the relaxation time onto A_{th} ? What determines the relaxation time?

Reply to Reviewers

Reviewer 1

In this manuscript, Dabelow and Reimann obtain a result for the behaviour of local observables in periodically-driven systems, in particular, that the heating (or generally the time evolution) of such observables slows down when the undriven system is close to a thermal state.

This is an important result by itself, and the existence of an analytical derivation makes this even more so--such results are not frequently possible in such systems. They also demonstrate its validity by numerical experiments--but see below for some questions on that.

Overall, given the amount of activity on the subject of Floquet systems, it is clear that this is an important and timely topic, and their result is likely to serve as a foundation for future work.

This manuscript does a good job overall presenting their analytical result. Their numerical demonstrations thereof are convincing as far as they go but, I think, incomplete. I therefore have a few questions before I feel I can recommend publication:

1. The numerical examples are at the fairly high temperatures $\beta=0.08, 0.2$ and 0.12 ; for all of these the observable at equilibrium is close to its infinite-temperature value. Can the authors discuss and show results from states where the temperature is lower? Is this technically difficult because the density of states at low temperatures is small for accessible system sizes, for example, or for some other reason? The authors should explain better these aspects of their work--either by presenting results with initial states with E (in their Eq. 5) further away from the middle of the spectrum, or discussing why that is not possible (or why their results do not apply to that regime, if they do not). At the moment all I can find about this in the main text is that the theory is valid for longer times at higher energies. Surely they have a more nuanced picture given all the work that they did, so why not present it?

We agree that the temperatures in our examples are relatively high. We have added further examples for lower temperatures in the new Supplementary Sec. S2.2 and discuss the temperature dependence of the effect there. We have also slightly expanded the relevant paragraph in the Methods (subsection "Discussion of applicability") and mention the additional examples in the main paper (section "Discussion and further examples").

Furthermore, we have added a short new paragraph above Eq. (6) in the main paper and the new Fig. S6 in the Supplementary Information, showing that an inverse temperature $\beta = 0.08$ (as chosen in our first numerical example) corresponds to a system energy which is still not that close to the energy at infinite temperature as the value 0.08 might possibly suggest at first glance. We have also added similar information in the caption of Fig. 2 and below Eq. (13).

As supposed by the referee, the reasons for our focus on relatively high temperatures are mostly of technical nature (weaker relative influence of heating, higher density of states, more homogeneous density of states). On the basis of the additional finite-size analysis in Sec. S2.3 of the Supplementary Information, we expect that increasing the system size will help to satisfy these properties for a broader range of temperatures, but we are currently bound by the available computing resources.

2. Do the authors have any results on the general behaviour of γ ?

For instance, is it monotonic for some classes of $v(s)$, or anything like that? (I realise it's not always monotonic--I'm simply asking whether they can show anything at all about this).

Given the great variety of possible driving protocols $f(t)$ and perturbation characteristics $\tilde{v}(E)$ and $v(t)$, respectively, it is quite hard to make general statements about the solutions $\gamma_\tau(t)$ of Eq. (10). Nevertheless, a few things can be inferred from the structure of Eq. (10) as well as from the alternative representations of $\gamma_\tau(t)$ in Eqs. (23) and (S29). We added a summary of the most important properties in Sec. S1.2 of the Supplementary Information and point to it in the main paper (towards the end of Section "Theory"). It complements the discussion of their characteristic time scales and amplitudes from Sec. S1.1 (previously Sec. S1).

As for the question about monotonicity in particular, we have never encountered an example where $|\gamma_t(t)|^2$ is monotonic for periodic $f(t)$, but cannot rule out either that there is a specific combination of $f(t)$ and $v(t)$ which achieves monotonicity. Therefore, we refrained from commenting on this issue in the paper.

I also have a few more minor comments:

1. Under Eq. 7, they say the average of the perturbation is over $E_{\mu-E_{\nu}} \approx E$ and not $|E_{\mu-E_{\nu}}| \approx E$. Is the distinction important? It seems to me that it is not. Is this correct?

This is absolutely right. We thank the referee for pointing this out, and we amended the paper accordingly.

2. Presentation-wise, the Methods sections do give useful information that can be pieced together to get a sense of how the main result of Eq. 9 would be derived, but the way they are presented leaves to the reader the task of putting them together. Instead of just presenting the pieces and letting the reader fit them together, perhaps a paragraph explaining how they fit could be added.

We recognize that the various ingredients of the main result were indeed laid out in a somewhat scattered fashion. To arrive at a more streamlined presentation, we have rewritten and extended parts of the Methods (most notably the subsection "Typicality framework") as well as the Supplementary Secs. S6–S8 (previously Secs. S4–S5).

3. Possibly not that important, but chronologically the first paper to show MBL is stable under driving is PRL 115, 030402 (2015) (it and Ref 20 appear to conclude similar things almost contemporaneously, but this is slightly earlier). This isn't really relevant to the correctness of their conclusions--this is just a comment.

We apologize for overlooking this fact and now cite the quoted work (as Ref. [38]) in those places where many-body localization and suppression of heating are discussed.

Reviewer 2

The paper develops a theoretical framework to describe the response of a quantum system, initially in an out of equilibrium state, subject to periodic driving. The main theoretical result is eq.(9) which predicts a particularly modest impact of driving, which can not significantly alter thermalization dynamics, as measured by some local observable.

Specifically, since γ in eq. (9) can not be larger than 1, the authors emphasize that the driving can not preclude or delay thermalization. This result is potentially interesting, and somewhat surprising. It might depend on the choice of driving operator V and the monitored observable A . Could it be that the modest impact of driving is due to small value of the 2pt function between A and V ? I guess if $A=V$ or more generally is large (here time evolution is by H_0), equation 9 with $|\gamma| \leq 1$ may not apply. I suggest authors investigate and comment on this point in the paper.

This is an important aspect that we probably did not stress sufficiently in the previous version of the manuscript: The adopted typicality framework cannot provide a reliable description for observables A that are strongly correlated with the driving operator V (for example $A = V$). The underlying reason is that the derivation uses randomized ensembles of V operators, but keeps A fixed, such that most V 's in an ensemble will not exhibit similarly strong correlations with A as the true V of interest. We added a corresponding remark in the ‘‘Theory’’ section, and we extended the relevant passage in ‘‘Discussion of applicability’’ in the Methods to point this out more clearly.

Regarding the question ‘‘Could it be that the modest impact of driving is due to small value of the 2pt function between A and V ?’’ we may remark that the absence of strong correlations between A and V does in fact not rule out a strong effect of the driving on the observable expectation values, as clearly demonstrated by our examples whenever the systems are away from equilibrium.

Furthermore, we have included additional numerics for the special case $A = V$ in the Supplementary Sec. S2.4. These results confirm that the theory is not suitable to describe such a situation. Nevertheless, we still observe a significantly suppressed reaction of the system close to equilibrium compared to the corresponding behavior away from equilibrium. Qualitatively, stalled response as the main effect we describe in the paper is thus observed beyond the conditions where the analytical theory can be formally applied. This is further supported by the simple heuristic argument provided in the ‘‘Conclusions’’ section of the main paper.

Another comment, to have predictive power, eq. (9) require function v from eq. (7) as an input. Can one define (7) through the Fourier of 2-point function of V , using arguments based on ETH? Without that, if one has to rely on direct diagonalization, the whole point of (9) disappears -- with the exact diagonalization one can calculate the dynamics of eq. (2) exactly. So, I suggest, if this is possibly to reformulate (9) solely in terms of observable quantities, e.g. $\langle V(t)V \rangle$ at temperature specified by ρ , etc.

Connecting $v(t)$ to the 2-point function of V is an intriguing suggestion and would indeed provide a deeper understanding of the response relation. Unfortunately, it seems that no such connection exists in general, but we can argue that $v(t) \simeq \langle V(t)V \rangle_{\rho_{\text{mc}}}/2$ at sufficiently high temperatures. We now mention this below Eq. (8) and discuss it in more detail in Supplementary Sec. S3. There, we also added a direct comparison between the numerically obtained dynamics and the theory using $v(t) = \langle V(t)V \rangle_{\rho_{\text{mc}}}/2$, yielding pretty good agreement again.

In any case, we emphasize that the predictive power of Eq. (9) does not rely on exact diagonalization of the system, even if the approximation $v(t) \simeq \langle V(t)V \rangle_{\rho_{\text{mc}}}/2$ is not applicable. First of all, it is not necessary to know all matrix elements $V_{\mu\nu}$ exactly to calculate the function $\tilde{v}(E)$ from (7). Instead, the coarse-grained matrix structure is sufficient (similarly as in the off-diagonal ETH), and we only need it in the initially occupied energy window. Moreover, one can often avoid assessing the matrix structure altogether and instead assume a rather generic form for $\tilde{v}(E)$, e.g., an exponential decay. The remaining scale parameters of such an ansatz can then be extracted from the response to a time-*independent* perturbation using the theory from Ref. [32]. In fact, this is precisely the

approach adopted in the examples from Figs. 1–3, i.e., we did not diagonalize any of those systems exactly.

Regarding “... the whole point of (9) disappears ...”, let us remark that, qualitatively, the prediction of stalled response as such actually follows from (9) even without any further knowledge of the function $v(t)$. In our opinion, this is an equally important point of our analytical theory. In the paper, this is now more explicitly emphasized at the beginning of Sec. “Discussion and further examples”.

An interesting situation suggested by the authors is when asymptotic value of $A(t)$, subject to undriven evolution, is different from thermal equilibrium. The authors considered such a case, but as I can judge from their Fig. 3, at late times their theoretical prediction (blue dashed line) is systematically different from the exact result (solid blue line). I suggest the authors comment on this point.

This systematic deviation is most likely caused by our truncation of the Magnus series at second order. We have adjusted the discussion of this example to clarify this issue, see the last paragraph of “Discussion and further examples.”

A few more comments about the manuscript. The authors called the modest response to driving “stalled” - I’m not sure why they choice this word and what it should convey. They also several times call their finding “astonishing.” I believe they should explain in the manuscript what they default expectation was which warrants this adjective.

We realized that our original description probably did not highlight the main characteristics of the “stalled response” effect sufficiently clearly. We hope that the revised introduction provides a much clearer picture from the beginning, and also explains why we think the effect is “astonishing”.

Apart from that, we find it difficult to understand why one may find, for instance, the numerical findings in Fig. 3 “not astonishing”: The same system exhibits a quite significant response to the periodic driving in one case (nonequilibrium initial conditions, blue and black), but does not show any visible reaction at all to the same driving in the other case (equilibrium initial conditions, green and red). We did our best to bring out this central point of our paper more clearly.

Regarding the name “stalled response” in particular, we will try to explain our choice in more detail, but must start by saying that neither of us has English as their first language, so we might be missing nuances in meaning, and we are open for alternative wording suggestions. As far as we understand, there are two general meanings of the verb “to stall:” (1) to bring to a standstill and (2) to delay or to defer (see, e.g., <https://www.merriam-webster.com/dictionary/stall>). Based on these two meanings, we believe that “the response is stalled near thermal equilibrium” is a rather concise summary of the main effects we observe: (1) For a system that is initially far from equilibrium, the observable response dies out and comes to a standstill as the unperturbed system thermalizes. (2) For a system that is in or close to equilibrium, the driving is barely noticeable on short-to-moderate time scales, and only manifests itself through heating on much larger times scales, i.e., the observable response is delayed.

Finally, they use blue lines (solid and dashed) in Fig.1, and this choice is very confusing, making the lines almost indistinguishable.

We adjusted the figures and now distinguish theory and numerics by color (as well as dashing), too.

Finally, in the beginning of the paper the authors mentioned recent NMR experiments as an example of systems where their theory applies. But toward the end they admitted that available information (about the experimental setting etc.) is not sufficient to make an accurate comparison, validating their theory. I therefore suggest to remove the mention of NMR from the introduction.

While we cannot quantitatively compare the experimental data to our theory, we still contend that the experiment clearly shows the essential qualitative signatures of stalled response as the main effect we describe. As is now pointed at the beginning of “Discussion and further examples”, one remarkable feature of our theory is the general prediction of stalled response *per se*, even in such a case where the available information is not sufficient for a quantitative comparison. In this sense, we still think that the experiment may be viewed as an important test of the theory.

Moreover, given the numerical evidence and the heuristic arguments laid out in the second paragraph of the “Conclusions” section, we believe that the effect itself may actually be observable even beyond the validity regime of the theory. Therefore, we think that it is fair to mention the experiment in the introduction. As a compromise, we have formulated the criticized passage a little more carefully and speak of “the main qualitative signatures of such a stalled response behavior” now.

Reviewer 3

This paper addresses the relaxation phenomena of quantum many-body systems under periodic driving fields. The manuscript is well organized.

The authors developed the theory of relaxation phenomena (Eqs. (9),(10),(11)) based on the Floquet-Magnus expansion up to the second order and argument with the typicality.

As a brief technical remark, we would like to emphasize that we adopt a Magnus expansion, but not a *Floquet*-Magnus expansion. That is to say, we do not use Floquet’s theorem to split the propagator into a periodic micromotion operator and a relaxation contribution from a time-independent Floquet Hamiltonian. While this is a technical detail, it should help to clarify why our results go beyond previous studies of driven systems based on Floquet theory. We have added a more detailed discussion of this technicality in Supplementary Sec. S4, and a less technical explanation of the same issue in the second paragraph of the Introduction.

The relaxation phenomena are governed mainly by the unperturbed Hamiltonian, and the values of initial relaxation go to the thermal equilibrium (equivalently to microcanonical) values given by the initial states. The periodic driving does not significantly contribute to the overall structure of the relaxation, which the authors dub stalled response. These observations are highly connected to the well-known Floquet prethermalization.

I could not understand how new these observations are at the moment.

As mentioned above, we tried to better clarify this issue by substantially extending the Introduction, by discussing the connection to Floquet theory in detail in the new Supplementary Sec. S4, and by amending the relevant passages in the main paper (notably the fifth paragraph and the second-last paragraph of “Discussion and further examples”). In a nutshell, and using the language of Floquet theory, our “stalled response” phenomenon means that the observable influence of the micromotion

operator becomes small near thermal equilibrium. To the best of our knowledge, such an insight cannot be found in the pertinent previous literature.

(1) So far, people have (presumably) thought that the prethermalization plateau is given by the Gibbs state with the Floquet Hamiltonian computed from the Floquet Magnus expansion, where the initial state gives the effective temperature. In this paper, the initial relaxation goes to the value of the thermal state of the unperturbed Hamiltonian that does not contain additional Floquet orders. A new aspect is that the prethermal-relaxation is dominated only by the unperturbed Hamiltonian.

We agree that the plateau of Floquet prethermalization is controlled by the Floquet Hamiltonian H_F within Floquet theory, whereas it is given by the unperturbed (or time-averaged) Hamiltonian H_0 in our case. The connection is established by observing that $H_F = H_0$ to lowest order (assuming that $f(t)$ averages to zero over one period) and by recalling that we truncate the Magnus expansion at second order. We understand that higher-order corrections could entail corrections to the plateau value, even though there are arguments that these corrections are typically small (see, for example, Ref. [24]). We have added a corresponding remark in the “Discussion of applicability” subsection of the Methods and also mention it in Supplementary Sec. S4.

(2) The authors repeatedly say that the stalled response is astonishing. However, the standard picture of Floquet prethermalization is that the state relaxes onto the thermalized state of Floquet Hamiltonian. Based on this picture, insensitivity to the driving field, starting from the thermal state, is already expected.

As far as we understand, the standard picture of Floquet prethermalization is based primarily on properties of the Floquet Hamiltonian, which only encodes the dynamics of the driven system at integer multiples of the driving period. From the Floquet Hamiltonian alone, no conclusions about the behavior between those discrete observation points can be drawn, i.e., observable expectation values can and will generally still exhibit non-negligible (and possibly very strong) periodic oscillations. The astonishing aspect of our result is that they do not do so if the undriven system is near thermal equilibrium.

We have rewritten the introduction as well as the pertinent discussion in “Discussion and further examples” to work out these important differences to standard Floquet theory more clearly.

(3) The sentence in page 4: ‘Our present result amounts to a highly nontrivial extension of this previously established insight. Namely, it implies that the plateau value is not only approached stroboscopically but continuously’.
I could not understand why the continuous approach is surprising, which is something expected. If this observation is highly nontrivial, the authors should explain the detail.

We hope that our answer to the previous points along with the reformulation of the criticized paragraph and the additional remarks in Supplementary Sec. S4 clarify why we see our theory as a nontrivial extension of Floquet prethermalization.

A particular example showing that one can have Floquet prethermalization (i.e., stroboscopically stationary dynamics) *without* continuous suppression of the response is Fig. 3 of the main paper. We recall that, in this case, the unperturbed system relaxes, but the state approached at long times differs from the thermal equilibrium state. We amended the text towards the end of the section

“Discussion and further examples” to highlight this aspect more clearly.

(4) Question: With your nice formula Eq.(9), can you estimate the relaxation time onto A_{th} ? What determines the relaxation time?

We are not entirely sure in what sense this question asks for an estimate of the relaxation time.

We can estimate the time scale in the sense that Eq. (9) predicts the driven system to relax towards A_{th} on the same time scale as the accompanying undriven system thermalizes. We have adjusted the discussion of the relevant time scales in Supplementary Sec. S1.1 to work this out more clearly.

If the question is asking for a concrete quantitative value of this time scale, then unfortunately no such estimate can be given within the present theory: Since our goal is to predict how the driving *changes* the system’s behavior, the undriven dynamics is an *input* to the theory. Therefore, no further information about its relaxation process can be extracted.

REVIEWER COMMENTS

Reviewer #1 (Remarks to the Author):

The authors have addressed all the points I had made in my review, and I have no further criticism. I am happy to recommend publication of the work as it stands.

Reviewer #2 (Remarks to the Author):

In my original report I raised a number of questions regarding the nature of the main result, its scope of applicability (and scope of applicability of the underlying theory), as well as various points concerning presentation. I would like to thank the authors who provided comments in their response and revised the paper. Below I will try to summarize how I assess the new version of the paper.

1. The paper puts forward an interesting observation that:

- i) periodic driving does not move the system out of equilibrium (for short and intermediate time scales)
- ii) for systems initially out of equilibrium, effect of periodic driving "stalls" (disappears) once the system finds itself in equilibrium state.

This observation is interesting and seems to be tested in a range of examples. To what extent the observation i) is new is less clear (this question is raised in other reports).

2. The scope of applicability of the proposed theory is unclear. The theory only applies if A and V are "orthogonal" but unless there is a symmetry which enforces that any two operators have an overlap. In case of a strong overlap, say $A=V$, theory breaks down, but as authors explain qualitative behavior remains. Does it mean "stalled response" is a completely universal phenomenon?

3. To have a predictive power, one needs to know function $v(t)$. In the example considered, the authors postulated $v(t)$ to be exponential. Why? More generally they explain that $v(t)$ is a Fourier of two-point function of $\langle V(t)PV \rangle$ where P is a project in the energy space. Two comments. First, does

the answer depend on details of the projector (shape of the filter). Second, one can express $v(t)$ as a convolution of $\langle V(t) \rangle$ and the Fourier of P . Say, if P is Gaussian filter with variance σ , then $v(t) \sim \int dt' C(t-t') e^{-\frac{\sigma^2}{2} (t')^2}$

4. It seems very natural to understand theoretical answer, eq. 9, in two natural limits. First, driving $f(t)$ has very small amplitude a . In this case one would expect some linear response theory to be applicable. Can one justify stalled response in this regime, and perhaps link it to some previously known phenomena? Second, one can T to be very large, reducing the problem to that one of sudden quench. What is the stalled response phenomenology in this regime?

5. Unfortunately the revised manuscript is not very clearly written. In the abstract stalled response is described as absence of driving effect for the system in equilibrium (in contrast to sizable driving effects for systems out of equilibrium). I don't think this is what authors try to convey with the word "stalled". They presumably mean that starting from out of equilibrium, the response stalls (disappears) once the system comes close to equilibrium.

6. In justifying importance of their result, the authors mention recent NMR experiment and claim it exhibits stalled response phenomenology. They even discuss in details one of the figures from that paper. In this case I think that figure should be added to this manuscript.

To conclude, to warrant publication in Nature Communications, I think the paper should make a clear case the phenomenon they find is novel and "astonishing" (in their own words). Current version did not convince me of that. I also strongly suggest to improve the theoretical part by addressing the points I raised above.

Reviewer #3 (Remarks to the Author):

The authors replied to all my comments. In fact, I do not still understand why they think it to be astonishing the stalled state lasts even during the stroboscopic times. The Floquet-prethermalization occurs extremely small period limit, which means that the system has no room to react to the fast-driving field within the period. So it is natural to expect that prethermalization occurs even for continuous time, while I agree with an opinion that this should be proven once we have such intuition. I suggest the author change the expression "astonishing" to a more moderate English word, because, I do not think that it is very astonishing. The paper is somewhat exaggerated.

However, it is important to prove the expected phenomenon, and the authors nicely derived an impressive compact formula, which is worth publishing in this journal.

Reply to Reviewer 2

We thank the referee for devoting his/her valuable time to the evaluation of our manuscript.

Before addressing the specific issues 1–6, we would like to make some general remarks.

In our opinion, the main point of the paper is the discovery of what we call the “stalled response effect”. To this end, we provide various numerical examples, an approximate analytical theory, and a simple qualitative explanation (now slightly extended upon revision). These three facets of the paper are of roughly equal importance in our own view.

The referee’s viewpoint seems to be rather different, putting the main emphasis on the following two questions: (i) Is the analytical theory sufficiently general? (ii) Does the theory admit quantitatively accurate predictions for some given example without too much numerical effort?

We repeat that these are not the actual main objectives of the paper. The predicted stalled response effect *per se* seems much more interesting to us than the quantitative details in particular examples. In this respect, the theory does a fairly good job, explaining the occurrence of the effect *per se* without requiring any further quantitative details of some specific model. As already mentioned by Reviewer 1, analytical predictions of this kind are quite rare and remarkable.

It is true that our theory has its limits of validity, but our numerics and qualitative explanations show that the stalled response effect itself persists beyond these limits. Of course, ultimately the effect itself also must have its limits, but a comprehensive exploration of this issue goes beyond our present scope. We think that the paper convincingly substantiates our main discovery in many different cases and already points out several key prerequisites like proximity to thermal equilibrium, suppression of heating, and moderate (not too small or large) driving amplitudes and periods (in particular, outside the linear response regime). Notwithstanding, we agree with the referee that studying the limitations in more detail remains as an important direction for future research.

Response to the issues 1–6:

1. We appreciate that the reviewer finds “observation (ii)” interesting. On the other hand, we think that when questioning the novelty of “observation (i)”, this should be substantiated by references to pertinent previous literature where such an observation can be found. We are not aware of such references.

The referee also says that “this question is raised in other reports”, while only Reviewer 3 actually raised such a concern in the first round, claiming that “stalled response” was essentially the same as “Floquet prethermalization”. This claim was refuted in our reply and, judging from the second report, Reviewer 3 seems to agree, but instead now raises the new objection that our systems have “no room to react to the fast-driving field”. However, this is once again refuted by our numerical finding that the black (unperturbed) and blue (perturbed) curves in Figs. 1–3 exhibit very notable differences for small-to-moderate times, i.e., the system evidently does have “room to react” as long as it is away from thermal(!) equilibrium.

2. As already said, it is not our ambition to show that stalled response is “a completely universal phenomenon”. Rather, we show that it can be observed in a considerable variety of cases, and we believe that this is already an interesting and new finding in itself.

More specifically, we are not sure what the reviewer might have in mind when saying “unless there is a symmetry which enforces that any two operators have an overlap”. The figures in the paper and the Supplementary Information provide various quite natural combinations of A and V for

which the theory works.

3. We numerically observed, for instance by exact diagonalization of smaller systems, that the function $\tilde{v}(E)$ (*not* the function $v(t)$!) can often be approximated quite well by an exponential function (see third paragraph of “Discussion and further examples” and Sec. S2.3), and that our theoretical prediction (9) does not depend very much on how exactly $\tilde{v}(E)$ is chosen in detail.

For the rest, the connection of $v(t)$ with the two-point function $\langle V(t)PV \rangle_{\rho_{mc}}$ is surely interesting and may merit further study. However, as stated above, it is not our main goal in this paper to come up with accurate quantitative predictions in particular examples. The comparison with the numerics only serves to show that the theory “works”. For the rest, we are not really interested in the detailed quantitative behavior of those particular examples. Nevertheless, we also point out ways to approximately determine $v(t)$ for anyone interested in adopting the theory quantitatively, and, again, finding additional ways may be an interesting direction for future research.

Regarding the proposed integral representation of $v(t)$, we were not able to understand the referee’s line of thought. The projector P mainly affects the statistics of the matrix elements of V . Why should this be equivalent to a temporal convolution/filtering? If anything, one would need to “Fourier transform” in the matrix indices, which would lead to a second, independent “time” argument for the “two-point function” and require substantial additional assumptions.

4. It is our impression that the reviewer is still looking for reasons why our present effect might not be new or unexpected. In the first round, “a small value of the 2pt function between A and V” was proposed as such a potential reason. By choosing $A = V$, we numerically disproved this suspicion.

In the second round, the reviewer is now asking for more details in the limits of small driving amplitudes and/or frequencies, apparently again for the purpose of gaining some intuitive understanding of the basic physics behind our effect (“link it to some previously known phenomena”).

This is a natural idea and we pursued this route of having a look at limiting cases ourselves already. Unfortunately, it did not help us to improve our understanding of the phenomenon. In fact, Fig. 1 already illustrates how the effect becomes weaker for very small or very large driving amplitudes or periods, and we had already discussed these limits in the last three paragraphs of the Section “Phenomenology”. More thoughts regarding the interplay of time scales and amplitudes can also be found in Sec. S1.1 of the Supplementary Information. Regarding the issue of linear response (small amplitudes), in particular, we also elaborate on it in Sec. S2.1 there.

To sum up, it is mandatory to focus on moderate driving amplitudes and periods. This is exactly why it is challenging to obtain an analytical theory (even a not “completely universal” one). In particular, it seems likely that only some “non-systematic” (non-perturbative) approach, such as our present random matrix methods, will do the job. It is well-known that the range of validity of such non-systematic theories (similarly as in the case of, for instance, density functional theory, renormalization and regularization techniques, or random matrix methods in general) is hard to predict *a priori*. Nevertheless, they are quite commonly considered as interesting and relevant.

5. We have amended the paper along these lines.

6. We have considered this suggestion, but feel that such a reproduction of several previously published figures is rather uncommon, and would entail a disproportionate text overhead, given that this experiment is discussed only in a single paragraph at the moment. The figures in question will be accessible to the readers by means of two or three clicks (starting out from our published online paper). Moreover, including the figures would also require to explain what experimental quantities are plotted, and hence what has been done in the experiment. We believe that all this is not really mandatory in our present paper and leave the issue to the editor to call.